# Understanding the feasibility of home-based rehabilitation in kidney transplant recipients: A mixed-methods interpretation

Roseanne E. Billany[1,2,3]*, Hannah M. L. Young[2,4,5], Courtney J. Lightfoot[1,2,3], Nicolette C. Bishop[6], Alice C. Smith[7], Matthew P. M. Graham-Brown[1,2,3]

1 Division of Cardiovascular Sciences, University of Leicester, Leicester, United Kingdom, 2 NIHR Leicester Biomedical Research Centre, University of Leicester and University Hospitals of Leicester NHS Trust, Leicester, United Kingdom, 3 Leicester Partnership for Kidney Health Research, University of Leicester, Leicester, United Kingdom, 4 Diabetes Research Centre, College of Life Sciences, University of Leicester, Leicester, United Kingdom, 5 Therapy Department, University of Hospitals of Leicester NHS Trust, Leicester, United Kingdom, 6 School of Sport, Exercise and Health Sciences, Loughborough University, Loughborough, United Kingdom, 7 Division of Public Health and Epidemiology, University of Leicester, Leicester, United Kingdom

* r.billany@leicester.ac.uk

## Abstract

### Background

The ECSERT randomised controlled feasibility trial reported that a structured home-based exercise programme was feasible in kidney transplant recipients (KTRs). This study presents a qualitative evaluation of the trial and programme of exercise. The results are integrated and interpreted via a mixed methods analysis.

### Methods

Fifty stable KTRs > 1-year post-transplant were randomised 1:1 to a 12-week home-based exercise programme or guideline-directed care control. Participants from the ECSERT trial intervention group ($n = 12$) completed individual semi-structured interviews immediately and 3 months after the 12-week exercise programme. Interviews were analysed using reflexive thematic analysis. Quantitative data on feasibility (recruitment, adherence, and attrition) and qualitative data were collected concurrently, and combined using convergent mixed methods analysis with a joint display to further explore the primary feasibility findings.

### Results

The mixed methods analysis facilitated the explanation and exploration of the successful feasibility findings. A key factor in recruitment was initial contact from a trusted healthcare professional. The benefits of additional health monitoring and the option to complete the programme later surpassed the disappointment of allocation to

**Data availability statement:** The data supporting the findings of this study are not publicly available due to privacy and confidentiality agreements with participants. Interview transcripts contain personal information, and while anonymised quotations may be used in publications, full transcripts cannot be shared to ensure participants cannot be identified. Researchers interested in accessing anonymised excerpts will be required to comply with ethical and confidentiality requirements and may contact the corresponding author or the Research Ethics Committee: East Midlands Nottingham 2 - REC Reference 19/EM/0290. Sponsor contact for queries: Cat Taylor; RGOsponsor@le.ac.uk.

**Funding:** The ECSERT study was funded by a project grant from Kidney Research UK (ref: KS_RP_003_20180913. HMLY is funded by an NIHR Advanced Fellowship [NIHR302926].

**Competing interests:** The authors have declared that no competing interests exist.

the control group. Key facilitators to exercise session completion were seeing change and preserving health, getting into a routine and the flexibility of home-based exercise, the variety of options, the importance of progression, the obligation to complete/being monitored, and the importance of support. At 3 months, many participants maintained some physical activity but described challenges with motivation, time, and illness.

## Conclusion

Elements of the ECSERT trial design, trial procedures, and exercise programme that contributed to the successful feasibility of the main trial were identified. These results will allow researchers and practitioners to maximise the effectiveness of future trials and programmes of home-based exercise in KTRs.

## Introduction

Access to specialist rehabilitation and physical activity support is recommended for people living with chronic kidney disease (CKD), including those who have received a kidney transplant [1,2]. Despite this, support is not routinely provided within the United Kingdom or elsewhere [3,4]. Kidney transplant recipients (KTRs) remain relatively inactive [5], despite the desire to become more active if greater support was available [6]. Supervised exercise interventions in KTRs improve cardiorespiratory fitness and a variety of traditional and non-traditional risk factors for CVD [7], which are common complications for this population and result in excess morbidity and mortality [8]. However, these programmes are not realistically deliverable in the current financial climate and have not translated to clinical practice. Nor do they invoke maintained behaviour [9–11], which is potentially attributable to low levels of patient activation and the failure of such programmes to engender sustained lifestyle changes [12,13]. Few studies have explored solely home-based delivery of rehabilitation in KTRs [14,15], and no studies have qualitatively evaluated their feasibility and acceptability in this population.

The ECSERT study was a pilot randomised controlled feasibility trial of a structured 12-week home-based exercise programme in stable KTRs exploring the feasibility of the trial, trial procedures, and the exercise programme [16]. The recently published results support the feasibility of a structured, home-based exercise programme and the associated trial in this population. Specifically, *a-priori* recruitment, adherence, and retention thresholds were all exceeded. The groups were well matched and there was encouraging representation of female participants and participants from a non-white background [17].

As part of a process evaluation of the ECSERT trial, we completed semi-structured interviews with participants who completed the trial to further explore (1) the acceptability of the design and conduct of the trial, and (2) the acceptability of the ECSERT exercise programme including exercise enjoyment, difficulties encountered, perceived advantages and disadvantages of the programme, and subsequent

longer-term engagement with physical activity. We present these findings as an integrated mixed methods analysis to support the understanding of the primary feasibility findings of the main trial. We hope that these results will allow researchers and practitioners to maximise the effectiveness of future trials and programmes of home-based exercise in KTRs.

## Methods

Reporting for this study follows the consolidated criteria for reporting qualitative research (COREQ) guidance [18], and the relevant CONSORT extensions [19].

### Study design

ECSERT (a pilot randomised controlled trial of a structured, home-based Exercise programme on Cardiovascular StructurE and function in Renal Transplant recipients) was a prospective, randomised, open-label, blinded endpoint (PROBE) feasibility study performed at one renal unit in the United Kingdom (prospectively registered: ClinicalTrials.gov [NCT04123951]). The full trial protocol and primary outcome (feasibility) results are reported elsewhere [16,17]. In summary, 50 KTRs were randomised (1:1) to either; (1) a 12-week home-based combined aerobic and resistance exercise programme ($n = 25$) or; (2) control ($n = 25$; receiving guideline-directed care). Outcomes were assessed at baseline and after 12 weeks [16]. Ethical and regulatory approval was obtained from the East Midlands-Nottingham 2 research ethics committee and the Health Research Authority (Ref: 19/EM/0209). Participants were recruited between 09/03/2020 and 16/02/2023, and provided written informed consent in the presence of research trial staff.

This manuscript reports the feasibility results from the primary manuscript ($n = 50$) [17], alongside new data from the qualitative analysis of semi-structured interviews ($n = 12$) with intervention participants (i.e., those that undertook the 12-week structured home-based exercise intervention). Integrated mixed methods analyses was used to explore the primary findings to provide insights and suggestions for future definitive trials and programmes of exercise in KTRs.

### Participants and recruitment

Full eligibility criteria for the ECSERT trial have been described elsewhere [16]. In summary, stable adult KTR (>1 year) at increased cardiometabolic risk, willing and able to consent, and able to safely perform exercise training and testing were eligible.

Participants from the intervention group were invited (by post) to be interviewed by phone or video call (whilst the participant was at home), or in-person (in a private room on a hospital site) after completion of the 12-week exercise programme. The predetermined purposive sampling framework included age, sex, kidney function, and intervention adherence to ensure maximum variation [20]. The minimum target sample size was 12: suggested by several studies and a recent review, to be in line with 'typical' sample sizes for phenomenological studies to yield saturation (where the addition of more data does not yield additional insights or issues) [21–23]. Three months after completing the exercise programme, participants who completed an initial interview were contacted for an additional semi-structured one-to-one telephone interview to explore the impact of the programme, if any, on subsequent maintenance of lifestyle and exercise habits.

### Exercise programme and control

In brief, the 12-week, home-based, structured exercise programme included aerobic and resistance training (4–5 sessions in total per week). The aerobic component of the programme was walking, jogging, cycling, or similar, depending on resources available and participant preference. Participants were asked to complete 2–3 sessions per week using a rating of perceived of exertion (RPE) of 13–15 (somewhat hard-hard) for 20–30 min. The resistance component of the exercise programme included a combination of 6–8 exercises per session chosen by the participant from a pool of twelve exercises (to provide variety) targeting upper and lower body and core muscle groups, using free weights and/or resistance bands.

The chosen pool of exercises included: squat, hip abduction, lunge, calf-raise, side-lunge, bicep-curl, bent-over row, reverse-fly, lateral-raise, chest-press, side-bends, and standing trunk rotation. Participants were asked to complete 6–8 resistance exercises twice a week (but not on consecutive days to allow appropriate recovery). Initially, they were advised to complete 1–2 sets of 10 repetitions (at 60% 1-repetition maximum [RM]), gradually increasing to 3–6 sets of 10 repetitions with a minimum of 30 sec rest between sets. Whilst participants could do aerobic and resistance sessions on the same day if they wished, they were advised to have at least one day in between resistance sessions for recovery. Participants were provided with an exercise diary which included additional instructions, dumbbells and resistance bands, and access to educational and instructional videos. Participants received a telephone call from a member of the research team every two weeks in order to discuss progression of the exercise and address any issues. Participants continued to receive guideline-directed care as did all participants randomised to the control group.

## Outcome measures

At baseline and after 12 weeks participants completed the following outcome measures: cardiopulmonary exercise test (CPET), cardiac magnetic resonance imaging (MRI), physical function (timed-up-and-go [TUAG]; gait speed; sit-to-stand-60 [STS-60]), balance and postural stability, physical activity (via accelerometery), handgrip strength, lower limb strength (via isokinetic dynamometry), body composition (via bioelectrical impendence), muscle size (via MRI and ultrasound), blood and urine analysis, and self-reported questionnaires. Full details have previously been reported [16].

## Qualitative data collection

Interviews were conducted via telephone, video call, or in-person depending on participant preference and COVID-19 restrictions. Post-intervention interviews were conducted by one researcher independent to the ECSERT trial (HMLY) to reduce bias and encourage honest and open responses. The researcher kept a reflexive diary to record comments and thoughts during and after the interviews and during data analysis. The topic guide was predetermined but amended after two interviews by REB and HMLY after discussion about question flow (S1 File). Questions covered: (1) the acceptability of the design and conduct of the trial, and (2) the acceptability of the ECSERT exercise programme including exercise enjoyment, difficulties encountered, perceived advantages and disadvantages of the programme. Interviews were digitally recorded, anonymised, transcribed verbatim, and imported into NVivo, Version 20.6.1.1137 for Windows (QRS International, Massachusetts, USA) which was used as a data management tool.

## Qualitative analysis approach

This study was conducted using a phenomenological approach to explore the lived experience of participants. Reflexive thematic analysis was used to identify themes at a semantic level [24]. Two researchers independently coded two transcripts (REB and HMLY) followed by collaborative review to ensure consistent interpretation and improved credibility prior to REB conducting the rest of the analysis. REB and HMLY refined final theme names for consistency. Illustrative quotes are labelled with sex and age ('*' and '**' distinguish two participants of the same age). Code frequency counts were used to determine saturation; when the frequency of new codes diminished with few or no more codes identified. Where any new codes were identified in later transcripts, higher-order groupings (namely salient themes) where checked for relevance. If new codes were incorporated into existing salient themes, saturation criteria were satisfied.

## Research team and reflexivity

At the time of interviews REB (female) was a Clinical Trials Facilitator with a BSc (hons) in Sports Science and an MPhil in qualitative research. As REB was the main facilitator for the ECSERT trial, all interviews were carried out by HMLY (female) as participants were more likely to be open about their experiences. Similarly, REB conducted all aspects of the

trial including outcome measures and exercise programme monitoring, therefore would likely have some prior perceptions. HMLY was a Specialist Renal Research Physiotherapist and an NIHR Development and Skills Enhancement Fellow with BSc (Hons), MSc, and PhD qualifications. HMLY had extensive skills and experience in conducting qualitative research. Participants were only aware of HMLY's position as a researcher and physiotherapist and no prior relationship was established. HMLY was only aware that participants had taken part in the ECSERT exercise programme.

## Mixed methods analysis

A convergent mixed methods analysis was used [25], where quantitative and qualitative data collection and analyses occurred separately and in parallel before being integrated. This method allows the individual strengths of each method to prevail efficiently, and allows a comprehensive understanding of the research question. A joint display was developed to visually portray how the two datasets converged, diverged, or expanded [25,26].

# Results

## Quantitative (feasibility)

Quantitative feasibility data are reported in the primary manuscript [17]. To summarise, 171 patients were screened and 94 (55%) were eligible and invited to take part in the study. Fifty of those invited (53%) were recruited across 22 months of recruitment. Consented participant characteristics were: age $50 \pm 14$ years (INT $49 \pm 13$; CTR $51 \pm 15$), 23 male (INT 10; CTR 13), eGFR $59 \pm 19$ mL/min/1.73m$^2$ (INT $60 \pm 20$; CTR $61 \pm 21$), 35 White British (WB), 13 South Asian (SA), 1 Caribbean, and 1 Mixed ethnicity (INT 17 WB, 7 SA, 1 Mixed; CTR 18 WB, 6 SA, 1 Caribbean). Intervention participants ($n = 22$ completed) recorded an average of $4.4 \pm 1.4$ exercise sessions per week (aerobic $2.8 \pm 1.1$; strength $1.6 \pm 0.5$). Three participants withdrew from the intervention group (1 COVID-19 infection, 1 recurrent urine infections unrelated to the trial, 1 time/family circumstances) and one from the control group (lost to follow-up; 8% attrition). There were no serious adverse events reported.

## Qualitative

Fourteen ECSERT intervention group participants were approached, of which 12 agreed to complete qualitative interviews (both immediate post-intervention and at 3 months). Two participants declined to participate. Baseline characteristics are summarised in Table 1 and individual exercise programme adherence in S2 Table in S1 File. Interviews were completed between June 2021 and December 2022 with a median duration of 51 min (IQR 43–54). S3.1-S3.3 Tables in S1 File provide additional quotations for themes outlined in the coming sections.

## Trial design and procedures

Table 2 outlines the themes and related quotations for qualitative results relating to the trial design and procedures.

**Perceptions of recruitment and randomisation. Theme 1: Experiences of recruitment method:** Participants signified the importance of initial contact being from a trusted nephrologist or that members of the exercise/research team were heavily linked to a nephrologist. This reduced any concern that participation would harm their kidney transplant.

**Theme 2: COVID-19 influence:** Although the majority of participants expressed concerns about contracting an infection, they trusted the procedures that were put in place to mitigate any risks. Concerns became less prominent as vaccines were introduced. For some participants, COVID-19 was a reason to take part in the trial. Some participants expressed being 'bored' and 'fed-up' due to shielding at home and the trial being a relatively safe way of leaving the house. For others, weight gain and increased inactivity due to shielding and the need to find alternative ways of getting active were important drivers.

**Table 1. Baseline demographics and clinical characteristics of interview participants.**

|  | KTR (n = 12) |
|---|---|
| Gender (M:F) | 6:6 |
| Age (years) | 52.1 ± 11.9 |
| Ethnicity (n, %) |  |
| White British | 8 (66.7%) |
| Indian | 4 (33.3%) |
| eGFR (mL/min/1.73 m²) | 65.9 ± 19.1 |
| Time on KRT (years) | 6.1 ± 5.2 |
| Daily moderate PA (min) | 75.2 ± 42.2 |
| Co-morbidities (n, %) |  |
| Hypertension | 12 (100) |
| Type II Diabetes | 3 (25) |
| Hyperlipidaemia | 7 (58) |
| Heart condition | 0 (0) |

Abbreviations: KTR, kidney transplant recipients.

**Theme 3: Expectations of randomisation:** All but a small number of participants had a preference to be randomised into the exercise programme. However, they felt they would have been content with being in the control group, largely due to having the choice of doing the programme later on if they desired and the chance to have additional health outcomes measured.

**Reasons for taking part. Theme 1: Desire for physical benefits:** Many participants highlighted the desire to gain tangible physical benefits from taking part in the trial. The benefits that were expected and/or desired were weight and fat loss, building muscle strength, increasing fitness and stamina, and improving sleep regulation.

**Theme 2: Exploring exercise:** Participants discussed that they wanted to use the trial to explore different forms of exercise, particularly resistance training, and gain new ideas. Other participants expressed exercise as being something that they wanted to do but never really found the time to explore before the trial, particularly in a supported manner.

**Theme 3: Altruism:** Many participants described helping others, helping the researchers and clinicians, and the importance of research as main reasons for taking part in the trial. They expressed gratitude for receiving their transplant and discussed wanting to 'give something back' in the hope that the research would help others in a similar position in the future.

**Theme 4: Additional health monitoring:** Participants described completing the trial as a way of gaining additional health information. For example, the additional blood tests, the CPET, and the cardiac MRI were perceived as a free 'health MOT' and provided the participants with additional peace of mind that their health was stable.

**Outcome measure acceptability.** In general, participants described the assessment visits as positive and convenient when completed on one day. Two participants reported that immediate feedback with normative values reinforced the purpose of the trial. The majority of participants identified the CPET as the most useful and acceptable outcome measure due to being able to see tangible evidence of change. It also provided confidence to exercise independently due to being supervised by a clinician, despite mask and cycle ergometer seat discomfort. Being able to test their fitness and see tangible differences was also a reason they favoured the STS-60. Gait speed and TUAG were considered the least important tests due to ease and limited ability to see change. This may be due to the ability and younger age of this population. Although participants described the questionnaires as necessary, the most common words to describe the process were: "long-winded, tedious, repetitive and difficult". Participants reported varying degrees of accuracy when

**Table 2. Themes relating to the ECSERT trial design and procedures.**

| Theme | Example quotations |
|---|---|
| **Perceptions of recruitment and randomisation** | |
| Experiences of recruitment method | "...the fact that it's people connected with your nephrologist [research team] that, suggesting you do it, give you the confidence that you're not going to do yourself a mischief which is good." (Male, aged 47) |
| COVID-19 influence | "I think with the whole Covid stuff just obviously I was at home, I was putting on a few extra pounds and I thought yeah actually this could be the kick I need to get myself into a much better shape which I need to be in, just from my kidneys' perspective and for health benefits." (Male, aged 26) |
| Expectations of randomisation | "I would say that I would have liked to have been in the exercise group but I wasn't sort of like dead set on if I'm not in the exercise I'm not going to take part, sort of thing. I was just like OK, yeah, that's fine, I can have another check-up in 12 weeks' time, sort of thing." (Male, aged 26) |
| **Reasons for taking part** | |
| Desire for physical benefits | "I was hoping to get out of, you know, muscles building and then a bit of belly fat, I wanted to lose a little bit of belly fat!" (Female, aged 49) |
| Exploring exercise | "I was interested to know if doing weights really did have any benefit because that would have been interesting to know if all these years I've not been interested in doing it, whether it actually could have benefited me." (Female, aged 44) |
| Altruism | "So it was a bonus but it was also the research as well which is needed for the kidney clinic as well you know, because there's changes all the time going on [with health], and anything that helps you know. Just the littlest things like that can help people in the future." (Male, aged 62) |
| Additional health monitoring | "I really liked it. I always like going – it's basically getting a free personal trainer effectively for a bit, having a bit of investigation to make sure you know, you might find health problems with the MRI thing at the beginning, and just getting confirmation that my heart's not about to explode. And then any excuse to make me a bit fitter I always quite like." (Male, aged 47) |
| **Outcome measure acceptability** | |
| | "I would say the one where – the bike ride [CPET] was the one where I could actually see physical improvements, so that's the one where I thought OK, well actually this is – because I knew what I felt and how I was afterwards on the initial one and that's the one where I could see a clear difference in OK, yeah, there has been a change in my stamina etc." (Male, aged 26) [re: CPET] |
| | "The bike because I do like my bike, I like going to the gym on the exercise bike, I do try and hit targets and know when I've had a good time, so to speak, that was important and I did come home and share my feelings with the family on that." (Male, aged 52[#1]) [re: CPET] <br> "…being the MRI scan it didn't bother me at all but it was just the bit when they had, you know when your heart's beating fast like it's exercising, that's the only bit that worried me." (Female, aged 43) [re: MRI] |
| | "I was going to say the stand up, sit down one as well because anything that really takes you to the edge of your sort of fitness level seems useful, and both of those two did, because at the end of, even though it was just a minute of stand up, sit down, you're absolutely [shattered] by the end of it. So yeah, both of those two I think." (Male, aged 47) [re: STS-60] |
| | "Oh the walking up and down to a chair and back, you walk three and a half paces, turn round, walk three and a half paces back and sit down. That one in three, I couldn't see that one…just the walking up and down seemed strange." (Male, aged 65) [re: TUAG] |
| | "Yeah, OK. I mean, they were quite longwinded but they have to be done and you have plenty of time, they don't sort of thrust it on you and say you need to do this now." (Female, aged 72) [re: questionnaires] |

Abbreviations: CPET, cardiopulmonary exercise test; MRI, magnetic resonance imaging; STS, sit to stand; TUAG, timed up and go.

completing the questionnaires, especially when they questioned their relevance. One participant described not reading the questions in detail, whereas another described putting aside specific time to answer them thoroughly. Participants expressed conflicting opinions about the cardiac MRI. Some found that it gave them the most 'peace of mind', whilst others described it as 'scary', unexpected, and a cause of claustrophobia. The cannulas were described as uncomfortable and the sensation of the heart rate increasing being unpleasant when the pharmacological stress agent (adenosine) was administered.

## Exercise programme

**Acceptability of the exercise aids and programme delivery.** Table 3 outlines the example quotations for qualitative results relating to the acceptability of the exercise aids and programme delivery.

The majority of participants highlighted that the exercise diary was useful and straightforward to complete and was enough to understand what was required for the exercise programme. One participant liked that the diary felt personal (e.g., individual starting weights for each resistance exercise were given during the instructional session). Other participants reported that being able to track progress, and make adjustments over time was encouraging. Participants described the resistance exercise videos as a useful supplement to the exercise diary, helping to confirm that they were performing exercises correctly. Some participants described the rating of perceived exertion (RPE) scale as informative to gauge how hard they were working and others felt that it was subjective and vague. Participants found the instructional session (performed via video call after the baseline assessment visit) helpful when combined with the exercise diary. Some participants would have preferred to have someone present to correct exercise technique. Other participants felt the presence of someone may have helped them feel motivated to exercise. Two participants did not like exercising at home, preferring to go elsewhere as it provided them with an obligation to attend.

**Barriers to exercise session completion.** Table 4 outlines the themes and related example quotations for qualitative results relating to the barriers to exercise session completion.

**Theme 1: Finding the motivation, making the personal commitment:** Many participants described the biggest barrier to completing the exercise sessions as not having the personal motivation. This was particularly referenced in relation to the resistance exercises, which they found the most challenging. Some also felt that there were other distractions at home, including family obligations.

**Theme 2: Frequent illness and other health conditions:** There were a multitude of conditions and ailments described which included musculoskeletal pain and discomfort, swelling, infections, and breathing problems. These put limitations largely on the intensity of the exercise that participants could perform and, in some cases, reduced the number of resistance exercises that were appropriate from the pool of 12. A number of participants who had fistulas in place from previous dialysis therapy, highlighted a fear of lifting heavy weights. Almost all participants reported either an illness or multiple illnesses, a hospital procedure, or hospitalisation during their 12-week intervention period. Whilst these in themselves resulted in missed exercise sessions, participants described 'getting back into' the exercise as challenging as they could not always restart where they paused. Many needed to make adjustments to their programme. This was described

**Table 3. Acceptability of the exercise aids and programme delivery.**

| Component | Example quotations |
| --- | --- |
| Exercise diary | "I directly went on a 2 kilogram and then I used to – every time – about like – every week I have improved sets – I start with two sets and then improving myself – so because of the diary I knew that from next week how many sets I have to improve and I want to push myself, because of the diary I was pushing myself as well." (Female, aged 49) |
| Instructional videos | "They were very good and they assisted in showing me for example which way to hold the weights and which way round you went, yeah, so they were very instructive." (Female, aged 58) |
| RPE scale | "…when we were doing the study at the hospital at the beginning, when we were doing the assessment so then obviously that gave me an idea of how you know like, how we should work as well. So if I knew that I was sweating and getting out of breath, I knew I was like on a 15, like a 14/15 so that's how I used to work it out, what part I was on." (Female, aged 43) |
| Exercise sessions (home-based delivery) | "No not really. No. The less visits to the hospital were great, especially if you're highly vulnerable as well and you pick up all sorts [of illness] in there so it was good that we did most of it from home." (Male, aged 62) |

Abbreviations: RPE, rating of perceived exertion.

**Table 4. Barriers and facilitators to exercise session completion and maintenance of activity.**

| Theme | Example quotations |
|---|---|
| **Barriers to exercise session completion** | |
| Finding the motivation, making the personal commitment | "…just the motivation to do it at home, like I said before, I found difficult, just because of what's going on in the background as well, so that's another reason I put the TV on just to give me that bit of focus, because sometimes if I'm doing stuff and the kids come in and they start asking you questions you're quite tempted to stop aren't you, so they always come in, it's like when you're on the phone they'll always find a way to interrupt…but I wouldn't say that was a major problem, I'd probably say the motivation to do that side was more difficult than any other outside factors." (Female, aged 44) |
| Frequent illness and other health conditions | "Only twice I got poorly where I had to, like the first time I was poorly I carried on…I didn't do the weights but I carried on doing the exercise bike but then the second time I felt poorly for a whole week I couldn't do anything and it was during half term and when I went back, you know it was hard to like build it back up again from what I'd already, you know I did that hard work the first like few weeks and then it was hard like trying to get back into it but I think like it's hard when you get like within this, it's hard when you get poorly because that really messes up the programme, like you can't have a full 12 weeks." (Female, aged 43) |
| **Facilitators to exercise session completion** | |
| Seeing change and preserving health | "Oh look at me doing sixteen reps instead of ten, oh god, yeah, no, it's the motivation that I'm preserving what health I have, promoting a better outcome are the reasons why I do exercise, otherwise I would not bother, if I didn't have to, I wouldn't." (Female, aged 58) |
| Getting into a routine and the flexibility of home-based exercise | "I kept routine, like, I used to do most of the exercise like weight lifting, I done it in the morning…I can do it whenever in the day [but] I don't think I would have done it then because once I start my housework then one after the other, I'm not working or anything, but then I knew that I have to keep one time that will suit me." (Female, aged 49) |
| Variety of options | "I would do four on one day and then four on another but I would pick different ones every time – so I did every one of them but, you know, on different days..." (Female, aged 72) |
| The importance of progression | "Yeah, I had a couple of times when I thought oh I know that takes me twenty minutes to do that, let me see if I can do it either faster, so do it in eighteen minutes, or go a bit further and it be a twenty five minute stint." (Female, aged 58) |
| Obligation to complete and being monitored | "Yeah pretty much like I didn't want to be like, oh this guy's been a bit lazy! He's got nothing to do, why is he not working out?" (Male, aged 26) |
| The importance of support | "Yes because every two weeks [clinical trials facilitator] was ringing me so like she was constantly there on the end of the phone asking me what am I doing, how am I doing you know, so she was a good support..." (Female, aged 43) |
| | "He didn't mind at all, if I was happy to do it he was happy and so he used to, you know, when I was doing my exercises he would be writing down..." (Female, aged 72) |
| **Maintenance of activity** | |
| Desire to Continue Exercise | "I'm going to repeat it purely because I want to prove to myself that I can be better because I really do want to get more healthy, so kindly [clinical trials facilitator] gave me another exercise book to restart and do that and I will do it but I want to start when I know that I can – to see if I can make a better difference." (Female, aged 55) |
| Continuation Methods | "I'm doing a bit more walking now. If I have to go down to the shop, we walk down there instead of taking the car. Most times. A few times due to time restrictions then we will take the car, but if we have a nice day and make sure the cat's not following us, we can sneak out and go down you know." (Male, aged 62) |
| Small but Maintainable Changes | "With regards to weights I feel they've been an addition to my life. I think I have persevered and they are accessible…at any time. So I have kept on with them. And my upper arms have improved a bit." (Female, aged 58) <br> "Yea treadmill and your book. I can do the book all now top to bottom. All exercises. I do it when I got time, half an hour…I'm doing treadmill now more than I used to when I was with your training. Back then I could do only 25 mins now I can do about 35 mins, 10 mins more." (Male, aged 52#2) |

*(Continued)*

**Table 4.** (Continued)

| Theme | Example quotations |
|---|---|
| Challenges of Continued Exercise | "Even now the programme's over and I promised myself that I'd carry on…because I knew I had to do it. I knew I had to do it, that's why I did it but now I don't have to do it. The programme's finished now and I said that once it's done, I will carry on but it's kind of like getting really hard now. I think it's mentally as well, thinking oh I'm not doing it any more so it doesn't matter if I don't go on the treadmill today, it doesn't matter if I don't go on the exercise bike today whereas like when I was doing the programme, regards even if I finished at 4 o'clock, I finished like doing my marking or whatever at 4, I said look I need to go to the gym." (Female, aged 43)<br>"I find silly reasons not to, like my husband might be around, so I'll think oh I'm not doing it while he's here. Or I'll think oh it's too near bed and that might be a bad thing because I might not sleep, or it's too early in the morning, you know, I find these reasons and then the moment goes and then you never get back to it." (Female, aged 58) |
| Barriers of Health and Time | "A feature of kidney patients lives really…They may have the best will in the world but certain times health takes over. The same as anyone really if they get a niggle or something but it takes us longer to recover and we get depressed in between that and it's a while to get back into the swing of things." (Female, aged 58) |

as frustrating and demotivating as were existing health conditions which affected the amount or type of exercise that they could do.

**Facilitators of exercise session completion.** Table 4 outlines the themes and related example quotations for qualitative results relating to the facilitators of exercise session completion.

**Theme 1: Seeing change and preserving health:** Seeing and experiencing positive physical changes inspired the most motivation to complete sessions and created a 'positive feedback loop', encouraging participants to see how far they could 'go'. Many participants reported improvements in general fitness which translated into the ability to complete day-to-day activities more easily. A number of participants referred to physical strength; feeling stronger than before the exercise programme. Some participants reported visible changes in muscle mass and others discussed having more stamina and feeling less tired during the day, even though they were doing more overall activity. Four participants highlighted that the programme had a positive impact on mental health which resulted in completion of more jobs around the house and being generally more active. Some participants reported weight loss, whilst others were disappointed their weight remained unchanged. The importance of changing both diet and exercise habits to induce a change in weight was highlighted. Some participants reported a greater overall awareness of their health and discussed the importance of being as healthy as possible after receiving the new kidney in order to enhance longevity.

**Theme 2: Getting into a routine and the flexibility of home-based exercise:** Participants discussed the ways in which they incorporated the activities into their daily life, and those who were most successful in completing sessions had ways of making the exercise fit into their existing routine or adjusting their routine around the programme. The flexibility of home-based exercise was discussed by many participants. Some also described not wanting to attend the hospital for exercise as they spent enough time at appointments.

**Theme 3: Variety of options:** Participants discussed that having options to choose from maintained interest, provided flexibility and alternatives if certain exercises were not suitable. They liked having a pool of resistance exercises to choose from so that they could vary their activity each time. Participants highlighted that it was beneficial for the aerobic exercise as they often did not have equipment. They could also vary their activities based on the weather. One participant did not share the same opinion; too much choice meant that they would opt for the easier exercises.

**Theme 4: The importance of progression:** Participants described that having a set plan in place was a motivating factor. This included having clear ways to progress the exercises which in turn provided positive feedback and motivation to continue. This structure was described as a way for them to 'keep on track' and know what they were doing was correct and not harmful.

**Theme 5: Obligation to complete and being monitored:** Participants discussed that being part of the trial was a motivational factor to complete the exercise sessions. Some felt a sense of obligation and some recognised that what they did would have an effect on the overall trial outcome. Others did not want to feel as though they had let themselves down by not completing sessions. Some participants described that 'being monitored' was a strong motivator. Even though the programme was remote, they knew that the exercise diaries would be seen and that they would have to redo the assessments.

**Theme 6: The importance of support:** Participants spoke positively about the phone calls that were received every two weeks to check on their progress. Some participants acknowledged that they were motivational and kept them on track with completing the exercise sessions; this was particularly pertinent to the exercise being home-based. All highly regarded being able to contact someone at all times; this was fundamental to the home-based delivery and to foster a feeling of support. They also described the support of family members aiding adherence to the programme, especially from acknowledgement, praise, and physical help (e.g., completing the diary with them). One participant reported that they had a protective family, particularly during the earlier stages of COVID-19 which restricted their ability to complete exercise outdoors.

**Maintenance of activity.** Table 4 outlines the themes and related example quotations for qualitative results relating to maintenance of physical activities.

**Theme 1: Desire to continue exercise and the act of continuing:** Almost all participants discussed a desire to continue exercise, whether it be continuing the programme, preferred aspects of the programme, taking up other exercise, or carrying on with prior activities in a modified way. This coincided with participants discussing being more 'health conscious' and more aware of their activity levels. Many participants reported taking a blank exercise diary to help them to continue and others reported having bought resistance bands/dumbbells.

**Theme 2: The challenges of continued exercise:** Whilst almost all participants described a desire to continue exercise after the programme, some were already struggling. The main reason was motivation, particularly without the structure and commitment of the trial and the lack of monitoring, both externally and personally. Time was also a barrier to continuing regular exercise. Some participants described finding alternative things to do, especially housework which acted as a 'good excuse' not to exercise. Two participants alluded to the challenges of having less time due to COVID-19 restrictions easing.

**Theme 3: Continuation methods:** Many participants reported already starting to continue with different forms of exercise such as repeating the exercise programme, continuing their favoured parts of the programme, or making small adjustments to their everyday activities. All participants outlined methods which would help them to maintain either the ECSERT programme of exercise or other exercise of interest in the longer term (S4 Table in S1 File).

Eleven participants completed a three-month telephone follow-up interview to explore exercise participation after the trial. One participant was uncontactable. The following two themes were derived from these calls.

**Theme 4: Small but maintainable changes:** Although almost all participants did not maintain the same level of exercise, most had maintained some aspects. Three participants had maintained some of the resistance exercises regularly and two participants had maintained them intermittently. Four participants were doing regular structured aerobic exercise. Some participants had noticed positive changes in weight, fitness, and strength. Others highlighted that although they were not doing the programme as they previously were, they had managed to maintain the elements that best fitted around their daily routines.

**Theme 5: The barriers of health and time:** The challenges of continuing exercise expressed by participants in the post 12-week interviews were reflected at 3 months, but the main focus was on health conditions and illness and the restriction of time. One participant who started off continuing exercise at a high level after the programme had a sudden illness that had left them unable to exercise in the six weeks prior to the telephone call which led to frustration. This was a common theme amongst participants who expressed that this was part of being a KTR. Many participants reported that

time was the biggest barrier to continuing the exercise at the same level. They discussed that when other events health happen, exercise becomes less of a priority.

## Mixed methods analysis

The mixed methods analysis facilitated the understanding of the recently published primary feasibility findings [17]. Table 5 shows the joint display of the qualitative and quantitative datasets. The ECSERT trial and exercise programme were shown to be feasible, meeting *a-priori* recruitment, retention, and adherence criteria. The qualitative dataset provides complementary reasoning for successful recruitment, reasons for rates of attrition and supportive information for the successful programme adherence, alongside potential barriers and suggestions for the maintenance of long-term engagement. Whilst outcome measure completion was quantitatively high, qualitative data presents an indication of the least and most important measures to participants. A summary of suggestions for progression to a definitive trial are presented in Table 6.

## Discussion

This work increases the understanding of the elements that resulted in the feasibility of the ECSERT trial in KTRs. Key themes contributing to the successful completion of exercise sessions were seeing change and preserving health, getting into a routine and the flexibility of home-based exercise, the variety of options, the importance of progression, the obligation to complete/being monitored, and the importance of support. Self-motivation and frequent illness and symptoms were key barriers. Importantly, the data shows some continuation of the exercise programme elements three months after trial involvement ended. It provides insights into elements that may support longer-term engagement, such as repeated testing, yearly reviews, and telephone check-ins, in order to encourage sustained physical activity behaviours.

### Trial design and procedures

Participants stressed the importance of the initial contact being from a clinician which provided the security that the trial was linked to a trusted person which likely led to the successful recruitment rates. The perception that clinicians, and more specifically a patient's own clinician, are most successful in recruiting to clinical trials has been previously explored in several populations. Qualitative interviews conducted in clinical trials teams revealed that recruitment was more successful in trials where a clinician had mentioned the study prior to the research nurse providing more details, even if the study was observational [27]. From the perspective of KTRs and trials of exercise, the importance of clinicians' approval is seemingly fundamental across all modes of exercise [6,28,29]. Many KTRs would not partake in exercise programmes or trials without the approval of their clinician to ensure they will not harm their transplant.

Outcome measure completion was high and the qualitative results provide valuable insight and further detail into the participant perspectives. The CPET is one of the most common measures to be used in exercise trials of solid organ transplant recipients (SOTR) [30,31], and calls have been made to consider functional exercise capacity tests that are more relevant to daily life and are less costly and resource-intensive [31]. However, in the present study, whilst completion was the lowest (but still high) for this outcome, participants described it as the most beneficial. It provided reassurance prior to exercising independently at home and tangible results where progress could be monitored post-intervention. The latter was an important facilitator for exercise maintenance and continuation. Whilst the perception of the benefits of exercise can sometimes be enough to motivate continuation, the inclusion of understandable and interpretable measures can support these perceptions [32]. There was discordance between the high questionnaire completion rates and the qualitative perspectives. Future studies should consider which questionnaires are the most important, valid, and reliable to measure the desired outcomes. This will prevent overburdening the participants and avoid inaccurate completion. There have been several calls for a core outcomes set of standardised outcome measures for exercise and physical activity research in both SOTR [31] and patients living with CKD and kidney transplants specifically [33]. Care should be taken to

**Table 5. Joint display with feasibility and qualitative results.**

| | Feasibility criteria | Feasibility results | Qualitative results | Mixed-method interpretation |
|---|---|---|---|---|
| **Eligibility** | X | **55% patients eligible** | X Patients not involved in screening process | N/A |
| **Recruitment** | Success of 20% of eligible subjects and ≥ 2 participants per month | **53% eligible patients recruited** Reasons for decline: - time/too much going - not interested - uncontactable - awaiting/receiving medical treatment - language Good representation of females and non-White British participants | - More likely to take part with consultant contact first - COVID-19 likely positive influence Reasons for taking part: - Desire for physical benefits - Exploring exercise - Altruism - Additional health monitoring | Qualitative results provide complementary data to explore the reasons for successful recruitment |
| **Attrition** | ≤30% | **8%** 1 due to COVID-19 infection; 1 due to recurrent urine infections unrelated to the trial; 1 due to time and family circumstances; 1 uncontactable | X No participants who withdrew consented to interview Randomisation to control acceptable due to: - Offer of exercise programme after trial - The benefits of additional health monitoring | Whilst withdrawals were not explored, qualitative data provides complementary evidence that randomisation was not a problem for participants |
| **Programme adherence** | Average of 3 exercise sessions per week recorded | **4.3±1.5 exercise sessions per week** | Barriers: - Motivation - Time - Frequent illness and getting back into exercise - Other medical conditions Facilitators: - Seeing change - Preserving health - Getting into a routine - Flexibility of home-based exercise - Variety of exercise options - The importance of progression - Obligation to complete - Being monitored - The importance of support A range of suggests for supporting long-term adherence: - Group sessions - Yearly reviews - Equipment provided - Supportive phone calls | Qualitative results provide supplementary information to explain the barriers and facilitators to exercise session completion and provide information for future programmes to adapt and support long-term engagement |
| **Programme compliance** | X | **Aerobic** RPE: 13 (target 13–15) Duration of sessions: 48±24 min (Target 20–30 min) **Resistance** Sessions per week: 1.6±0.5 (Target 2) Exercises per session: 7 (Target 6–8) Number of sets of 10: 1.8 week 1 & 3.5 week 12 (Target 1–2 increasing to 3–6) | | |
| **Outcome measure acceptability** | X | **Missing data ≤10% for all data apart from:** Cardiac MRI: 12% baseline, 13% follow-up CPET: 12% baseline, 11% follow-up IPOS-renal: 14% baseline, 13% follow-up Accelerometer: 11% follow-up Urine samples: 11% follow-up | - Number of measures acceptable - CPET and STS-60 identified as most useful - Cardiac MRI provided the most peace of mind but was the most unpleasant for many - Gait speed and TUAG least useful or at least not explained well - Clear and prompt feedback on how results relate to 'normal' values expected/desired - Questionnaires most notably described as long-winded, tedious, repetitive, and difficult - Some participants did not necessarily focus on accurate questionnaire completion | Qualitative results provide supplementary information on the most useful measures to participants and the need for clear feedback. Some indication that questionnaire data may not be accurately completed |

Abbreviations: CPET, cardiopulmonary exercise test; MRI, magnetic resonance imaging; STS, sit to stand; TUAG, timed up and go.

**Table 6. Summary of suggestion for moving forward to a definitive trial.**

| | Description of problem or highlight of positive factors | Trial/Programme issue | Solutions within definitive trial |
|---|---|---|---|
| Sample size | Need for a sufficiently powered primary outcome | Trial | • Example power calculation based on VO$_{2peak}$ mL/kg/min as the primary outcome indicates a total sample size of 186 participants |
| Screening | Eligibility exceeding or equal to other trials but there is room for improvement | Trial | • Revise inclusion/exclusion criteria for the exercise intervention<br>• Conduct a multi-centre trial |
| Recruitment | Recruitment rates good and exceeding *a-priori* criteria<br>Good representation of female and non-White British population, however the latter could still be improved | Trial | • Clinician to approach initially<br>• Provide brief summary sheet prior to detailed PIS<br>• Involve family (or preferred persons) in recruitment<br>• Sub-study for interviewing decliners<br>• Consider information in other languages<br>• Keep outcome measures relevant to participants as gaining more information on health was a reason for participation |
| | | Programme | • Delivery based on individual preferences<br>• Initial in-person sessions if required |
| Retention | Extremely low attrition rates (being mindful of the +ve influence of COVID-19) | Trial | • Longer trial follow-up period to allow for illness/procedures<br>• Consider trial design to include attention control/sham control |
| | | Programme | • Offer a range of delivery (some in-person sessions if needed)<br>• Support getting back into exercise after illness<br>• Longer programme length and follow-up period |
| Randomisation | Evidence of good distribution between groups apart from body weight | Trial | • Consider stratification |
| Blinding | Participants, investigators, and outcome assessors not blinded | Trial | • Provided provisions for outcome assessor blinding<br>• Participant and investigator blinding not possible |
| Fidelity | Not measures in the current trial | Trial | • Manual of protocol<br>• Training for deliverers<br>• Continued monitoring possibly with modified TIDier checklist |
| Outcomes | All outcome completion good but qualitative data provided detail on patient preferences<br>Evidence of a mismatch between quantitative and qualitative data | Trial | • Provide detailed explanations of measures<br>• Consider measures that were considered useful to participants: CPET and STS-60<br>• Include more objective measures<br>• Include only the most relevant questionnaires<br>• Revise measures based on exercise (e.g., present study did not measure upper body strength but participants favoured the upper body exercises<br>• Pick measures based around activities of daily living as these were important to participants<br>• Outcomes to measure maintenance post-intervention |

Abbreviations: CPET, cardiopulmonary exercise test; STS, sit to stand.

seek the opinions of KTRs directly in the development of these outcome sets, as research and patient priorities can differ substantially.

## Exercise programme

Every participant valued the exercise diary as it allowed a quick and efficient way of documenting progression week-by-week. Lahham et al. concluded that a diary was an acceptable and valid way to ascertain exercise participation after comparing with an objective measure during the final week of their 8-week home-based exercise programme in pulmonary rehabilitation [34]. A systematic review of 14 exercise trials in people with various health conditions investigated exercise adherence and outcomes when instructions were provided using a multimedia approach compared with verbal and written [35]. They concluded that multimedia approaches to exercise instruction may increase exercise adherence but evidence

was insufficient to determine whether there was any impact on outcomes [35]. The qualitative results in the current study support the use of multimedia approaches to exercise instruction as participants found the additional instructional videos helpful to aid participation.

There were polarised views regarding instructional sessions and/or the exercise programme including some supervised (in-person) elements. Opinions were clearer for the instructional session(s) with many participants believing that seeing the exercises live and reinforcing correct technique would be beneficial. Although in many cases, the videos, diaries, and phone calls were sufficient without any in-person elements. Concern during the programme about physical ability was the only neutral response received within the patient satisfaction questionnaire (previously published [17]) which could be addressed by in-person instructional sessions to build confidence. This is supported by previous qualitative work in which KTRs with less confidence and experience of exercise desired supervised exercise sessions to build confidence; this was not required long-term [6].

A key barrier to exercise session completion was illness and/or symptoms of comorbidities [17], which is further explored in this qualitative work. Participants reported difficulty with restarting exercise after a bout of illness or hospital procedure, particularly as the sessions were home-based and fear of negative consequences was high. Given the frequent nature of illness in this population, programmes should include methods that help participants regain pre-illness levels of exercise. This may be in the form of an in-person session(s) to rebuild confidence in ability and reduce worry. Similarly, comorbid conditions, particularly musculoskeletal, were troublesome. Participants made use of the flexibility of the programme, particularly the bigger pool of resistance exercises to choose from to mitigate some effects. Future studies should continue to provide adaptable and varied exercise to avoid discomfort whilst exercising. Whilst less comorbidities have been identified as being predictive of exercise adherence in home-based studies [36], some of the difficulties explained by participants point to a potential need for education or direction by their healthcare professionals. For example, fear of lifting weights with the fistula arm can lead to overprotection and underuse and restriction to daily activities which can reduced QoL [37]. A systematic review suggested that arm exercise training actually promotes arteriovenous fistula (AVF) clinical and ultrasonographic maturation [38]. Similar fears were expressed around exercise with osteoporosis.

Participants who observed changes reported being more motivated to continue exercising. For participants lacking in self-motivation, as discussed, initial in-person sessions or some in-person training sessions may be a way to encourage self-motivation during home-based exercise programmes. Participants reported improvements in strength, stamina, fatigue, and mental health and seemed to place the highest value on improvements that impacted daily life, such as less fatigue and ability to work longer. Some of the most potent reasons for exercise participation have been previously identified as maintaining independence [39] and increasing daily activity performance [40].

Low self-motivation was one of the biggest challenges to completing the exercise sessions, particularly as they were largely self-directed. Lack of self-motivation was previously identified as a barrier to exercise in KTRs [6]. An umbrella review of 55 studies assessing the key factors associated with adherence to physical exercise in patients with chronic disease and older adults identified social support as a crucial factor in continued exercise [32]. Telephone calls every two weeks were viewed as motivational and necessary to continued home-based exercise participation. Regular communication has been identified as a factor that may influence physical activity in the short- and long-term [32]. In a study of 104 adults with chronic health conditions, those who received gym-based exercise and home-based exercise supplemented with telephone support achieved similar outcomes [41]. Whilst telephone calls require additional resources, they may have contributed to the high exercise adherence in the current study and these (or similar approaches) appear fundamental to the success of home-based programmes. Similarly, participants spoke of the motivational benefits of familial support. Bachmann et al. identified studies whereby low support were associated with lower home-based exercise adherence [42]. Future home-based programmes and trials of exercise may consider how family members and telephone communications might be utilised in a sustainable clinical setting as useful methods of support.

Getting into a routine, the flexibility of home-based exercise, and the variety of options were important factors for exercise session completion. Collado-Mateo et al. identified that integration into daily living was strongly associated with higher activity adherence [32]. Whilst many factors are at play in the transformation of physical activity into a lifestyle habit, the authors highlighted that their background and the preferences of participants were the most crucial, and only programmes that are in line with these can be successful.

### Long-term engagement with exercise

Many participants had continued with some elements of the programme, usually the ones which they enjoyed the most, and/or ones that they had integrated into their daily routine. The importance of routine was discussed as a facilitator to exercise session completion, but it appears to be an important factor during the transition to longer-term engagement. A Cochrane review comparing home- and centre-based programmes in older adults found that participants were more likely to adhere to home-based programmes in the longer-term compared to centre-based [43]. In trials with a follow-up of ≥ 12 months exploring exercise in patients with coronary heart disease, there was a small but significant difference in exercise capacity in favour of the home-based exercise [44]. This supports in-person sessions being supplementary to a predominantly home-based programme.

Suggestions by participants for encouraging long-term engagement with exercise after the programme included many of the elements of the facilitators of exercise session completion previously mentioned, such as monitoring of progress. Suggestions were repeated CPET testing, check-in phone calls, yearly reviews with repeated assessments, and continued support from someone close to the nephrology department with exercise experience (e.g., a physio or exercise physiologist). In a systematic review of 11 studies assessing interventions to achieve ongoing adherence for adults with chronic conditions following a supervised exercise programme, there were no significant differences in long-term adherence between intervention types (centre-based, home exercise with telephone follow-up, home exercise with no follow-up, and weaning-based programmes) [45]. This is perhaps indicative that longer-term engagement is entirely personal and future programmes may wish to include a multitude of options.

### Strengths and limitations

A key strength of this study is the mixed methods analyses, allowing for further explanation of the feasibility of the ECSERT trial design and procedures, and the exercise programme [17]. Equal importance of both datasets was ensured by the concurrent collection and analysis before integration [25,26]. Qualitative data collection by an independent researcher and reflexivity diaries supported qualitative rigour. As with most trials of physical activity, it is likely that the sample were somewhat already motivated to exercise and therefore the results may not be generalisable. Effort should be made to engage participants who do not participate in any form of physical activity as the applicability to this subgroup of the population remains uncertain. Some factors of the trial design could not be explored as participants from the control group were not interviewed. Finally, there is a lack of quantitative data on maintenance activity to explore whether qualitative findings aligned. This data was not collected due to limited resource.

### Clinical implications

The development and implementation of rehabilitation programmes within healthcare systems has been described as essential [46], yet implementation in CKD in particular is extremely low despite the inherent benefits to health and QoL [47]. The ECSERT study results provide clear recommendations for the delivery of home-based rehabilitation which can be adopted for different healthcare settings through the inclusion of key recommendations for programme components. A clear progressive programme with a variety of activity options and a method of documenting progress (written or multimedia) is essential. Support with integrating activity into daily routine supports long-term engagement, as does structured assessments, monitoring, and support from healthcare professionals. Instructional sessions (largely required in-person)

ensure a strong foundation to engage in activity in a home setting with confidence. Methods of supporting the return to activity after illness, procedures, or hospitalisation to prevent drop-off and return to inactivity should be considered an integral part of any programme. The inclusion of these components should be considered in the design and delivery of any home-based programme to encourage success.

## Conclusion

This qualitative evaluation of the ECSERT exercise programme and trial procedures provides rich data which can be used to shape future home-based programmes of exercise and the trials that evaluate them. This data supports and helps to explain the trial feasibility results and introduces programme benefits that were not reflected in the quantitative data alone. Importantly, the data shows some continuation of the exercise programme elements three months after trial involvement ended. It provides participant suggestions of longer-term programme elements, such as repeated testing, yearly reviews, and telephone check-ins in order to encourage sustained physical activity behaviours. These results will allow researchers and practitioners to maximise the effectiveness of future trials and programmes of home-based exercise in KTRs.

## Supporting information

**S1 File. Supplementary materials_V1.**
(DOCX)

## Acknowledgments

This is a summary of independent research funded by Kidney Research UK, and carried out at the National Institute for Health and Care Research (NIHR) Leicester Biomedical Research Centre (BRC). The views expressed are those of the author(s) and not necessarily those of the funders, the NIHR, or the Department of Health and Social Care. For the purpose of open access, the author has applied a Creative Commons Attribution (CC BY) licence to the Author Accepted Manuscript version arising from this submission.

## Author contributions

**Conceptualization:** Roseanne E. Billany, Nicolette C. Bishop, Alice C. Smith, Matthew P. M. Graham-Brown.

**Data curation:** Roseanne E. Billany, Hannah M. L. Young, Matthew P. M. Graham-Brown.

**Formal analysis:** Roseanne E. Billany, Hannah M. L. Young.

**Funding acquisition:** Matthew P. M. Graham-Brown.

**Investigation:** Roseanne E. Billany, Hannah M. L. Young, Matthew P. M. Graham-Brown.

**Methodology:** Roseanne E. Billany.

**Project administration:** Roseanne E. Billany.

**Software:** Roseanne E. Billany.

**Writing – original draft:** Roseanne E. Billany.

**Writing – review & editing:** Roseanne E. Billany, Hannah M. L. Young, Courtney J. Lightfoot, Nicolette C. Bishop, Alice C. Smith, Matthew P. M. Graham-Brown.

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
