## [Decision Letter · Decision Letter 0]

2 Oct 2025

Dear Dr. Billany,

Thank you for submitting your manuscript to PLOS ONE. After careful consideration, we feel that it has merit but does not fully meet PLOS ONE’s publication criteria as it currently stands. Therefore, we invite you to submit a revised version of the manuscript that addresses the points raised during the review process.

Reviewers were overall satisfied of the contribution, there are some minor suggestions to implement and reflect upon. As Editor I do agree that it could be nice to at least reflect in the text about these minor points from the Reviewers:

- Clarity on clinical implications – Suggest, in the final discussion, how the findings can be translated into concrete recommendations for home-based rehabilitation programs in different healthcare systems.

- Qualitative data – Although the justification for not making full transcripts available is valid, the authors could strengthen transparency by highlighting criteria for theoretical saturation and providing additional examples in the supplementary material.

- Generalizability – More clearly acknowledge that the findings may reflect a group already motivated to exercise, and that applicability to less engaged recipients still needs to be tested

- Adding more objective measures is highly recommended please add this to your recommendations, and also longer follow up period.

We look forward to receiving your revised manuscript.

Kind regards,

Simone Borsci, Ph.D.

Academic Editor

PLOS ONE

Journal Requirements:

“ Kidney Research UK”

4. We noted in your submission details that a portion of your manuscript may have been presented or published elsewhere.

“Feasibility results (previously published) are provided within the manuscript to provide the joint display of quantitative and qualitative (new) results. This prevents the reader having to refer back.”

5. We note that you have indicated that there are restrictions to data sharing for this study. For studies involving human research participant data or other sensitive data, we encourage authors to share de-identified or anonymized data. However, when data cannot be publicly shared for ethical reasons, we allow authors to make their data sets available upon request. For information on unacceptable data access restrictions, please see http://journals.plos.org/plosone/s/data-availability#loc-unacceptable-data-access-restrictions.

7. We note that there is identifying data in the Supporting Information file <ECSERT Qual Supplementary Material_V1.docx>. Due to the inclusion of these potentially identifying data, we have removed this file from your file inventory. Prior to sharing human research participant data, authors should consult with an ethics committee to ensure data are shared in accordance with participant consent and all applicable local laws.

-Location data

Please remove or anonymize all personal information (<Age and Participant ID>), ensure that the data shared are in accordance with participant consent, and re-upload a fully anonymized data set. Please note that spreadsheet columns with personal information must be removed and not hidden as all hidden columns will appear in the published file.

8. We note that this data set consists of interview transcripts. Can you please confirm that all participants gave consent for interview transcript to be published?

If they DID provide consent for these transcripts to be published, please also confirm that the transcripts do not contain any potentially identifying information (or let us know if the participants consented to having their personal details published and made publicly available). We consider the following details to be identifying information:

- Names, nicknames, and initials

- Age more specific than round numbers

- GPS coordinates, physical addresses, IP addresses, email addresses

- Information in small sample sizes (e.g. 40 students from X class in X year at X university)

- Specific dates (e.g. visit dates, interview dates)

- ID numbers

Or, if the participants DID NOT provide consent for these transcripts to be published:

- Provide a de-identified version of the data or excerpts of interview responses

- Provide information regarding how these transcripts can be accessed by researchers who meet the criteria for access to confidential data, including:

a) the grounds for restriction

b) the name of the ethics committee, Institutional Review Board, or third-party organization that is imposing sharing restrictions on the data

c) a non-author, institutional point of contact that is able to field data access queries, in the interest of maintaining long-term data accessibility.

d) Any relevant data set names, URLs, DOIs, etc. that an independent researcher would need in order to request your minimal data set.

For further information on sharing data that contains sensitive participant information, please see: https://journals.plos.org/plosone/s/data-availability#loc-human-research-participant-data-and-other-sensitive-data

If there are ethical, legal, or third-party restrictions upon your dataset, you must provide all of the following details (https://journals.plos.org/plosone/s/data-availability#loc-acceptable-data-access-restrictions):

1. A complete description of the dataset

2. The nature of the restrictions upon the data (ethical, legal, or owned by a third party) and the reasoning behind them

3. The full name of the body imposing the restrictions upon your dataset (ethics committee, institution, data access committee, etc)

4. If the data are owned by a third party, confirmation of whether the authors received any special privileges in accessing the data that other researchers would not have

5. Direct, non-author contact information (preferably email) for the body imposing the restrictions upon the data, to which data access requests can be sent

Reviewers' comments:

Reviewer's Responses to Questions

**Comments to the Author**

1. Is the manuscript technically sound, and do the data support the conclusions?

Reviewer #1: Yes

Reviewer #2: Yes

2. Has the statistical analysis been performed appropriately and rigorously?

Reviewer #1: Yes

Reviewer #2: Yes

3. Have the authors made all data underlying the findings in their manuscript fully available?

Reviewer #1: Yes

Reviewer #2: Yes

4. Is the manuscript presented in an intelligible fashion and written in standard English?

Reviewer #1: Yes

Reviewer #2: Yes

Reviewer #1: Clarity on clinical implications – Suggest, in the final discussion, how the findings can be translated into concrete recommendations for home-based rehabilitation programs in different healthcare systems.

Qualitative data – Although the justification for not making full transcripts available is valid, the authors could strengthen transparency by highlighting criteria for theoretical saturation and providing additional examples in the supplementary material.

Generalizability – More clearly acknowledge that the findings may reflect a group already motivated to exercise, and that applicability to less engaged recipients still needs to be tested.

Reviewer #2: Thanks for submitting your valuable manuscript, important topic very good and detailed methodology

Adding more objective measures is highly recommended please add this to your recommendations ,also longer follow up period .

**Do you want your identity to be public for this peer review?** For information about this choice, including consent withdrawal, please see our Privacy Policy

Reviewer #1: No

Reviewer #2: **Yes: ** Seham Elazab

---

## [Author Response · Author response to Decision Letter 1]

17 Oct 2025

Please see point by point response attached. Many thanks.

---

## [Decision Letter · Decision Letter 1]

28 Oct 2025

Understanding the feasibility of home-based rehabilitation in kidney transplant recipients: a mixed-methods interpretation

PONE-D-25-40533R1

Dear Dr. Billany,

We’re pleased to inform you that your manuscript has been judged scientifically suitable for publication and will be formally accepted for publication once it meets all outstanding technical requirements.

Kind regards,

Simone Borsci, Ph.D.

Academic Editor

PLOS ONE

Additional Editor Comments (optional):

Reviewers' comments:

Reviewer's Responses to Questions

**Comments to the Author**

Reviewer #1: All comments have been addressed

2. Is the manuscript technically sound, and do the data support the conclusions?

Reviewer #1: Yes

3. Has the statistical analysis been performed appropriately and rigorously?

Reviewer #1: Yes

4. Have the authors made all data underlying the findings in their manuscript fully available?

Reviewer #1: Yes

5. Is the manuscript presented in an intelligible fashion and written in standard English?

Reviewer #1: Yes

Reviewer #1: The revised version represents a clear improvement over the previous submission. The manuscript is now technically sound, logically structured, and written with clarity. The methodology is well detailed, and the discussion appropriately contextualizes the findings within current literature.

**Do you want your identity to be public for this peer review?** For information about this choice, including consent withdrawal, please see our Privacy Policy

Reviewer #1: No

---

## [Editor Report · Acceptance letter]

PONE-D-25-40533R1

PLOS One

Dear Dr. Billany,

I'm pleased to inform you that your manuscript has been deemed suitable for publication in PLOS One. Congratulations! Your manuscript is now being handed over to our production team.

Kind regards,

on behalf of

Dr. Simone Borsci

Academic Editor

PLOS One